# The Luminous Fungi of Japan

**DOI:** 10.3390/jof9060615

**Published:** 2023-05-26

**Authors:** Yuichi Oba, Kentaro Hosaka

**Affiliations:** 1Department of Environmental Biology, Chubu University, Kasugai 487-8501, Aichi, Japan; 2Department of Botany, National Museum of Nature and Science, Tsukuba 305-0005, Ibaraki, Japan

**Keywords:** bioluminescence, fungi, mushroom, Japan

## Abstract

Luminous fungi have long attracted public attention in Japan, from old folklore and fiction to current tourism, children’s toys, games, and picture books. At present, 25 species of luminous fungi have been discovered in Japan, which correspond to approximately one-fourth of the globally recognized species. This species richness is arguably due to the abundant presence of mycophiles looking to find new mushroom species and a tradition of night-time activities, such as firefly watching, in Japan. Bioluminescence, a field of bioscience focused on luminous organisms, has long been studied by many Japanese researchers, including the biochemistry and chemistry of luminous fungi. A Japanese Nobel Prize winner, Osamu Shimomura (1928–2018), primarily focused on the bioluminescence system of luminous fungi in the latter part of his life, and total elucidation of the mechanism was finally accomplished by an international research team with representatives from Russia, Brazil, and Japan in 2018. In this review, we focused on multiple aspects related to luminous fungi of Japan, including myth, taxonomy, and modern sciences.

## 1. Introduction

The occurrence of dim-glowing mycelia on fallen twigs, which was first recognized as unknown ‘shining wood’, and luminous mushrooms on rotten trees has fascinated people around the world (Figure 1A), and Japan is no exception. The special interest in bioluminescent fungi in Japan is probably related to the richness of fungal diversity, which has led to a love of mushrooms and mushroom consumption in this country. In addition, a tradition of night-time activities could also be a factor.

The climate of mainland Japan (Hokkaido, Honshu, Shikoku, and Kyushu) ranges from subarctic in the northern part to temperate in the southern part; it is typically characterized by a cold winter and humid summer. The peripheral Izu Islands, Bonin Islands (Ogasawara Islands), and Ryukyu Archipelago (including Amami and Okinawa Islands) have subtropical oceanic climates with mild winters and hot humid summers. Because of wide-ranging climates with high humidity and species richness in woods and mountainous areas, Japan has a high biodiversity of fungi. Currently, approximately 13,000 described species of fungi have been reported from Japan, with possibly even more undescribed species [1]. Mycophagy has been popular in Japanese food culture, probably since Japan’s Jomon Period (–10th BCE) [2]. Various species of mushrooms have been cultivated and are always available on the market, such as *Lentinula edodes* (*Shii-také*, in Japanese), *Flammulina filiformis* (*Enoki-také*, formerly recognized as *F. velutipes*), *Hypsizygus marmoreus* (*Buna-shimeji*, also known as *H. tessulatus*), and *Pholiota microspora* (*Nameko*), which are indispensable for everyday Japanese cuisine [2,3]. Many amateur mycologists have been devoted to the understanding of Japanese fungal diversity, and several new mushroom species are found and described every year all across Japan by amateur and professional Japanese mycologists.

Since the old days, especially after the Edo Period, when people had more free time to enjoy their lives, there have been Japanese traditions of enjoying nature at night-time, including watching the moon (*Tsukimi*) and fireflies (*Hotaru-gari*), listening to insect calls (*Mushi-kiki*), and night hiking (*K*ō*ch*ū*-tozan*) [4]. Thus, it is not surprising that people accidentally witnessed bioluminescent mushrooms or glowing mycelia in the dark, and sometimes this glowing was thought to be caused by *Y*ō*kai*: creatures, presences, or phenomena that could be described as mysterious or eerie [5]. In Japan, many terrestrial bioluminescent organisms have often been discovered by ordinary people or amateur naturalists based on sporadic observations. Examples of such organisms include the bioluminescent earthworms *Microscolex phosphoreus* [6] and *Pontodrilus litoralis* [7], the millipede *Paraspirobolus lucifugus* [8], the springtail *Lobella* sp. [9], the tiny mushroom *Marasmiellus lucidus* [10], and the scarlet mushroom *Cruentomycena orientalis* [11].

In this paper, we attempt to review bioluminescent mushrooms in Japan and related topics, such as folklore and taxonomy, and some recent research results on bioluminescent fungi in Japan. For this purpose, we intentionally cite many studies written in Japanese to share hidden achievements from Japan with the world.

## 2. Folklores

There are many folkloric stories about strange glows around the world [12], including Japan. One example is *Kitsune-bi*, meaning ‘fox’s fire’; it is very interesting coincidence that old Japanese called an unknown glow on the ground ‘fox’s fire’, using the same metaphor as Europeans calling glows ‘foxfire’. Another is *Mino-bi*, meaning ‘raincoat fire’; *Mino* is a Japanese traditional raincoat made out of rice straw, *bi* is fire or glow, and a folklore about the mysterious glow of wet *Mino* was called *Mino-bi*. Both *Kitsune-bi* and *Mino-bi* are partly considered to be responsible for the luminous mycelia growing on rotten wood or the straw of *Mino* [13,14]. *Mino-bi* was sometimes believed to be the work of foxes [15]. Figure 1B is a masterpiece *Ukiyo-é* woodblock by Hiroshige Utagawa at the end of Japan’s Edo Period, showing foxes bearing fire in their mouths under a large Japanese hackberry tree. Of note, Sakyo Kanda (1874–1939), the author of the book “*Shiranui, Hitodama, and Kitsune-bi*”, which sheds light on the mysterious luminescence, was a biologist of luminous organisms, and he considered the major cause of the Japanese foxfire to be a misinterpretation of people’s lanterns [14].

The glowing of old pine trees was known as the mysterious phenomena of *Hikari-matsu* (luminescent pine tree [16]; the reference originally written in 1765) or *Ry*ū*-t*ō** (dragon’s lantern [17]), and these supernatural phenomena can most likely be explained by the luminescence of *Armillaria*’s mycelia [18]. *Armillaria* are known to be one of the common pathogens that cause wood rot in *Pinus* pine trees (*Armillaria* root rot, *Narataké-by*ō** in Japanese), which does occur in Japan [19].

Until recently (approximately 50 years ago), people on Hachijo Island (an island located in the Izu Islands) called the glow of the luminous mushrooms in the woods *Hato-no-hi*, meaning the fire of pigeons [20]. On this subtropical island, more than seven bioluminescent mushroom species are distributed, of which *Mycena lux-coeli* and *Mycena chlorophos* exhibit especially strong glows [21]. The Japanese wood pigeon *Columba janthina* is also distributed in the island woods. Most likely, people made a connection between the unidentifiable glow of luminous mushrooms in the night-time forest and the unsettling night call of the wood pigeon [22,23].

## 3. The Tale of the Bamboo Cutter

It is believed that luminescence of mycelia appeared as early as in a description by Aristotle (4th BCE) [12,24]. This roughly corresponds to the end of the Jomon Period in Japan, when people in mainland Japan were primarily hunter–gatherers and no written records were available. The first appearance of luminous fungi in Japanese literature was in ancient tales in Japan’s Heian Period (6–12th century).

The Tale of the Bamboo Cutter, “*Taketori Monogatari*”, was written by an unknown Japanese author in the late 9th or early 10th century and is recognized as the oldest Japanese work of fiction [25,26]. In the first part of this tale, there is an impressive scene in which a man called the Old Bamboo Cutter finds a glowing bamboo in the field.

“One day he noticed among the bamboos a stalk that glowed at the base”(translated by Keene, 1998 [25])

When he examines it, he finds a lovely little girl approximately three inches tall, named Shining Princess (*Kaguya-himé* in Japanese). She grows to be an adult, and she refuses several proposals of marriage from noble men, including the Emperor, and finally returns to the moon. As it turns out, she was a princess from the Moon.

What was causing the bamboo to glow? A Japanologist, Katsumi Masuda (1923–2010), hypothesized that the cause of the glowing was the luminous fungus *Panellus pusillus* (*Suzume-také* in Japanese) [27], and a historian, Michihisa Hotate, agreed with that idea [26]. This species also grows on the rotten culm (ringed stem) of Japanese bamboo [28,29]. The distribution records of *P. pusillus* were mainly in the southern part of Japan: Bonin Isls., Matsuyama (Ehime Prefecture) [28], and Hachijo Isl. [22,30], but also in central Japan [28]. Shidei reported the growth of *P. pusillus* on bamboo in Kyoto [29]. As the story is probably set in Kyoto, and at that time many bamboo craftworkers originating from the southern Kyushu region worked there [26], it can be speculated that the author of the tale had learned of the phenomenon of bamboo glowing from the craftworkers’ experiences, inspiring the famous opening scene [27]. *Mycena chlorophos* (*Yak*ō*-také* in Japanese) is another candidate; it is distributed mainly in the southern islands, such as Hachijo Isl. and Bonin Isls. [30,31], but is also found in various localities in mainland Japan (Honshu: from Kanto west to Kyushu, including Kyoto) [31,32], and it sometimes grows on bamboo. Of course, we do not exclude the possibility of the glow being caused by luminous mycelia of unknown identity on rotten bamboo [14] (Figure 2) or nonfungal organisms, such as the princess firefly *Luciola parvula* (*Himé-botaru*), which sometimes appears in bamboo groves [33,34].

## 4. Current Commercialization

Currently, the phenomenon of fungi bioluminescence is familiar to many people in Japan. For example, there are the characters of the Pokémon Card Game (a card game that appeared in Japan in 1996 and is now a worldwide success) (Figure 1C), capsule toys (small toys in vending machines packaged in a plastic capsule) (Figure 1D), and picture books that focus on luminous mushrooms [22,35,36] (Figure 1E). TV programs featuring mushrooms often include the topics of luminescence of some mushroom species. Artificial cultivation methods of *M. chlorophos* have been established [37,38], and the species has been used for special exhibitions in several local museums and botanical gardens (e.g., Hachijo Visitor Center, Tokyo; Yumenoshima Tropical Greenhouse Dome, Tokyo; and Nagoya City Science Museum, Aichi). A culture kit is available online [39]. Night-time hiking ecotours to watch glowing *M. chlorophos* are one of the most economically significant tour activities in Hachijo Isl. and Bonin Isls. [22,40], and trips to watch glowing *M. lux-coeli* have occasionally been held at various localities in southern Japan, such as the Amami Islands (Kagoshima Prefecture), Mt. Yokokura (Kochi Prefecture), and Ukui Peninsula (Wakayama Prefecture).

## 5. Taxonomy

### 5.1. Bioluminescent Species in Japan

In Japan, scientific surveys of fungi started around the 18th century when Japanese scholars were emancipated from Chinese herbalism *Honz*ō*-gaku* and began genuine native studies on Japanese flora. For example, a Japanese herbalism scholar, Tomohiro Ichioka (1739–1808), compiled a monograph of local fungi, “*Shin-you Kinpu,”* in 1799 and mentioned (probably) *Omphalotus japonicus* as *Kumahira* with illustration and the remarks “luminescence at night and poisonous.” Another Japanese herbalism scholar, Konen Sakamoto (1800–1853), compiled a monograph of the Japanese fungi “*Kinpu”* in 1835 and described *O. japonicus* as “*Tsukiyo-také*” with illustration and the remarks “this mushroom is called *Tsukiyo-také* because of luminescence at night”. Though based on pre-Linnean classification, these are probably the earliest scientific references about the bioluminescent mushroom in Japan. However, other luminous species were not described until the 20th century. This is partially because of the climate diversity of Japan. Mainland Japan, where most Japanese people including scholars are located, is characterized by a subarctic to temperate climate, while many luminous mushroom species, especially of the *Mycena* group, are distributed in tropical and subtropical regions. In other words, *O. japonicus* is the only bioluminescent mushroom species commonly (frequently) observed in Japan.

Indeed, *O. japonicus* is the first luminous mushroom species described scientifically under the Linnean system, which was formulated in 1915 by a mycologist, Seiichi Kawamura (1881–1946) [41]. The second piece of scientific evidence of luminescent fungi from Japan was reported by Yosio Kobayasi (1907–1993), who reported the luminescence of four known (currently three) species: *Favolaschia peziziformis, Panellus pusillus, Mycena chlorophos* (from Bonin Isl.), and *Mycena cyanophos* (=*M. chlorophos*) (Bonin Isl., and also from Hachijo Isl.) [42].

Before and during the Second World War, a Japanese researcher of bioluminescent organisms, Yata Haneda (1907–1995), extensively surveyed luminous mushrooms when he stayed at Palao Tropical Biological Station in Palau as a researcher under the mandate of Japan (during 1937–1942) and as Army Civil Administrator of Shonan Museum (the present National Museum of Singapore) in Singapore (during 1942–1945). After the Second World War, he returned to Japan and continued his luminous mushroom survey at Hachijo Island, Japan, and described several luminous mushrooms from Japan with assistance from the British mycologist/botanist Edred John Henry Corner (1906–1996) [43]. Of note, at the end of the Second World War, Corner was a captive prisoner of Japan. Thus, the official relationship between Haneda and Corner was that of enemies, but they struck up a scientific friendship during and after the war [44]. Although many of these species names described by Haneda and Corner were invalidly published, which unfortunately caused taxonomic confusion [45,46], their contributions paved the way for understanding the diversity of luminous mushrooms in Japan after the Second World War; *Mycena lux-coeli* (*Shiino-tomoshibi-také*) was collected by Haneda on Hachijo Island and described by Corner, and the species name remains valid.

Even recently, many new localities of luminescent mushroom species have been recorded, and Terashima and her colleagues described eight new luminescent species from southwestern Japan in their book [47]. Currently, approximately 100 species of luminous fungi have been recognized [48,49], of which 25 species are distributed in Japan [22,46,47].

In this section, all luminescent fungal species recognized in Japan are listed with remarks. Phylogenetic positions of these species are not presented in this paper, but some previous studies based on genome-scale DNA data have clearly demonstrated the relative positions of major bioluminescent genera and the polyphyly of bioluminescent taxa among mushroom-forming fungi [50,51]. Species that were “excluded, doubtful or insufficiently known” [46] were not included. Of note, *Nothopanus noctilucens* is sometimes listed as a luminescent species distributed in Japan [46]. However, the Japanese *Pleurotus noctilucens* (=*Nothopanus noctilicens*) *sensu* Inoko is an invalid name [52], and the true *Pleurotus noctilucens* Lév. (Syn. *Nothopanus noctilucens*) has not been reported in Japan [53]. *Mycena illuminans* has been reported as a luminescent species distributed in Japan [46]. However, this species is often considered a synonym of *M. chlorophos* [54] and thus is not included in the following list. The Japanese names were adopted from the list by Katumoto, 2010 [55], unless they have a more recent name.

#### 5.1.1. Family **Mycenaceae**

***Cruentomycena orientalis*** Har. Takah. & Taneyama

Japanese name: *Gahnetto-ochiba-také* [47]

Remarks: ‘*Gahnetto*’ means garnet in Japanese. The suffix ‘*-také*’ means mushroom. This species was described from Ishigaki Island, southern Japan [47]. The luminescence of the mycelium and fruitbody of this species and other similar species was reported in Fukuoka, Miyazaki, and Miyagi Prefectures [11,56,57]. The luminescence of the fruitbody was weak compared to that of the mycelium and detected only by a long-exposure CCD camera.

***Dictyopanus foliicola*** Kobayasi

Japanese name: *Konoha-suzume-také* [55]

Remarks: The Japanese *konoha* and *suzume* mean leaf and sparrow, respectively (‘sparrow’ represents a small creature in Japanese [58]). The mycelia and fruitbodies are luminous. This species has not been officially reported since the original description by Kobayasi from Miyazaki Prefecture [59]. The taxonomic status of this species warrants further study.

***Favolaschia peziziformis*** (Berk. & M. A. Curtis) Kuntze (Figure 3)

Japanese name: *Enashi-rasshi-také* [55]

Remarks: This species was originally described from the Bonin Islands but is also known on Hachijo Island, Okinawa and other countries in Australasia [60]. Whole fruitbodies are reported to be bioluminescent. *Enashi* means the lack of a stipe. *Rasshi* derives from the genus name *Laschia* in honor of German mycologist Wilhelm Gottfried Lasch (1787–1863) [58].

***Mycena chlorophos*** (Berk. & M. A. Curtis) Sacc. (Figure 4)

Japanese name: *Yak*ō*-také* [55]

Remarks: The Japanese *Yak*ō*-také* means ‘night-illuminating mushroom’. This species is distributed in Honshu (probably from Aomori, the northernmost prefecture [61]), Shikoku, Kyushu, Izu Islands, and Bonin Islands [62]. In addition, the species is widely recorded in the Southern Pacific islands, e.g., Polynesia and Micronesia [62]. This species is listed in the Japanese Red Data as endangered in Fukushima, Chiba, and Miyazaki Prefectures [63]. The bioluminescence of the fruitbody is considered brighter than many other known luminous mushrooms, but some strains, such as a strain in Miyazaki and Aomori Prefectures, seem darker compared to those in Hachijo and Bonin Islands [10,36]. The draft genome sequence of this species (Hachijo Isl. strain) has been assembled [64]. We consider *Mycena cyanophos* (Berk. & M.A. Curtis) Sacc. to be a synonym.

***Mycena daisyogunensis*** Kobayasi

Japanese name: *Hyūga-yak*ō*-také* [55]

Remarks: This species was collected from Daisyogun Cave in Miyazaki Prefecture in Kyushu (*Hyūga* is an old name of Miyazaki Prefecture) [59], but no further collections have been made since the original description. The taxonomic status of this species warrants further study.

***Mycena flammifera*** Har. Takah. & Taneyama

Japanese name: *Mori-no-ayashi-bi* [47]

Remarks: The Japanese *Morino-ayashi-bi* means ‘forest ghost-fire’. This species was described from Ishigaki Island, southern Japan [47]. The morphological differences from the better-known bioluminescent species, *M. manipularis* (Berk.) Sacc. are subtle, and the taxonomic status of this species warrants further study.

***Mycena lazulina*** Har. Takah., Taneyama, Terashima & Oba (Figure 5)

Japanese name: *Konruri-kyūban-také* [47]

Remarks: The Japanese *Konruri-kyūban-také* means ‘ultramarine-colored sucker mushroom’ because of the presence of a vivid blue (*Konruri*) disk-like (*Kyūban*) base. This species was described from the Yaeyama Islands, southern Japan [47]. Although its morphological characteristics seem to indicate that it belongs to the genus *Mycena*, the phylogenetic tree presented by Terashima et al. [47] has a very long branch leading to this species. The quality of DNA sequence data warrants further investigation.

***Mycena lux-coeli*** Corner (Figure 6)

Japanese names: *Shiino-tomoshibi-také*, *Hachijō-yakō-také* [55]

Remarks: The Japanese *Shiino-tomoshibi-také* means ‘*Castanopsis* tree’s lantern mushroom’. This species was originally described from Hachijo Island, but it is currently known from wider areas of central to southern Japan, mostly along the Pacific Ocean [60]. This species is listed in the Red Data as endangered in Mie Prefecture and as vulnerable in Miyazaki Prefecture [63].

***Mycena luxfoliata*** Har. Takah., Taneyama & Terashima

Japanese name: *Kareha-yak*ō*-také* [47]

Remarks: This species was described from the Ishigaki and Iriomote Islands, southern Japan [47]. Its bioluminescence was observed from mycelia on fallen leaves (*Kareha* means fallen leaves).

***Mycena manipularis*** (Berk.) Sacc. (Figure 7)

Syn. *Filoboletus manipularis* (Berk.) Singer, *Polyporus hanedae* Kawam.

Japanese name: *Ami-hikari-také* [55]

Remarks: The Japanese *Ami-hikari-také* means ‘reticulated luminous mushroom’. This species is known from central to southern Japan and has also been recorded in many other countries, including Indonesia and Australia [60]. It is listed in the Red Data as endangered in Chiba Prefecture and as near threatened in Miyazaki Prefecture [63]. The unique feature of this species is that its stems, rather than caps, are brightly luminous (Figure 7). The bioluminescent property seems erratic; it has been reported for the strain on Okinawa Island that nonluminescent and weak-luminescent fruitbodies sometimes appeared when cultivated in the laboratory [65]. The bioluminescence of the local strain in Miyazaki Prefecture seemed weaker [10]. Currently, the species is often called *Filoboletus manipularis* (Berk.) Singer.

***Mycena pseudostylobates*** Kobayasi

Japanese name: *Kyūbantaké-modoki* [55]

Remarks: The Japanese *Kyūbantaké-modoki* means ‘pseudo sucker-mushroom’. This species was recorded from Miyazaki Prefecture, but no definitive collections have been made since the original description by Kobayasi, 1951 [59]. The taxonomic status of this species warrants further study. The mycelium is bioluminescent, but the luminosity of the fruitbody is unknown [59].

***Mycena stellaris*** Har. Takah., Taneyama & A. Hadano (Figure 8)

Japanese name: *Hoshino-hikari-také* [47]

Remarks: The Japanese *Hoshino-hikari-také* means ‘starlight mushroom’. This species was described from the Ishigaki and Okinawa Islands, southern Japan [47]. The bioluminescence of the whole fruitbodies was recorded.

***Panellus pusillus*** (Pers. ex Lév.) Burds. & O. K. Mill. (Figure 9)

Japanese names: *Suzume-také*, *Hinano-uchiwa* [55]

Syn. *Panellus gloeocystidiatus* (Corner) Corner (Japanese name, *Suzume-také-modoki* [55])

Remarks: The Japanese name *Hinano-uchiwa* means ‘princess fan’. This species is known from central to southern Japan but is also widely reported from North and South America and Australasia [60,62]. It often grows on bamboo.

***Resiomycena fulgens*** Har. Takah., Taneyama & Oba (Figure 10)

Japanese name: *Ginga-také* [47]

Remarks: This species is known from Yaku Isl. (Kagoshima Prefecture), Hachijo Isl., and Kochi Prefecture [47]. The fruitbodies are small (up to ca. 3 mm), but they often grow in large numbers on the standing timber of *Castanopsis*, visually evoking an image of the Milky Way (*Ginga* means the Galaxy or Milky Way). Whole fruitbodies were reported to be bioluminescent [47].

***Roridomyces*** sp.

Japanese name: *Aya-hikari-také*

Remarks: Its taxonomic status has not been thoroughly studied, but it presumably represents a new species of the genus based on several morphological characteristics. Bioluminescence of Japanese samples (spores) was reported by Kurogi, 2015 [10]. The Japanese name *Aya* is derived from the fact that the species was discovered from Aya, Miyazaki Prefecture [10]. This species is listed in the Red Data as endangered in Miyazaki Prefecture [63].

#### 5.1.2. Family **Omphalotaceae**

***Marasmiellus lucidus*** Har. Takah., Taneyama & S. Kurogi

Japanese name: *Himé-hotaru-také* [47]

Remarks: *Hotaru* means firefly in Japanese. This species was discovered in Miyazaki Prefecture [47] during a survey of the *Himé-botaru* firefly (*L. parvula*) [10]. The whole fruitbodies were reported to be bioluminescent.

***Marasmiellus venosus*** Har. Takah., Taneyama & A. Hadano

Japanese name: *Himé-hikari-také* [47]

Remarks: The Japanese *Himé-hikari-také* means ‘princess luminous mushroom’. This species was described from Oita Prefecture in Kyushu [47]. The whole fruitbodies and mycelia are both reported to be bioluminescent. This and the previous species belong to the genus *Marasmiellus,* but their taxonomic treatment warrants further investigation. Currently, no other species are known to be bioluminescent in the genus *Marasmiellus,* and their accurate phylogenetic relationship to other bioluminescent species will give important insights into the evolution of bioluminescence in fungi.

***Omphalotus japonicus*** (Kawam.) Kirchm. & O. K. Mill. (Figure 11)

Syn. *Lampteromyces japonicus* (Kawam.) Sing.

Japanese name: *Tsukiyo-také* (old names: *Watari*, *Bunano-kataha*, *Kumahira*, *Hikari-goke*, and *Hotaru-také*) [55].

Remarks: The Japanese *Tsukiyo-také* means ‘moonlit-night mushroom’. This species is distributed widely in mainland Japan and is thus arguably one of the most well-known bioluminescent mushrooms in Japan. An anecdote in the mid-Edo Period (ca. 1800s), “*Zoku Sanshū Kidan*”, introduced a story called “*Nanao K*ō*rin*” where there was a bright luminescent mushroom called *Yamiyo-také,* meaning black-night mushroom, in Nanao (the current Nanao City in Ishikawa Prefecture); it claimed that the luminescence was strong enough to illuminate 1 m square when holding 2–3 pieces “like noon” [66].

Of course, this story most likely contains some hearsay exaggeration (the true luminescence of *O. japonicus* is such that “the fungi of different sizes could be easily recognized at a distance of thirty meters” in pitch dark, and of course not like noon, [41]), but this mushroom could possibly be *O. japonicus* because the story also introduces its gastrointestinal toxicity for humans, which is a characteristic property of this species [67]. The poisonousness of this species is well recognized in Japan because the fruitbody is similar to several Japanese edible species, including *Pleurotus ostreatus* (*Hira-také*, in Japanese), *Pleurotus pulmonarius* (*Usu-hirataké*), *Lentinula edodes* (*Shii-také*), and *Sarcomyxa edulis* (*Muki-také*), such that it is often consumed mistakenly [40,68]. In the 12th century tale “*Konjaku Monogatarishū*”, there is a story where a priest in Nara planned to kill his old supervisor to obtain the supervisor’s position by serving cooked *O. japonicus* (old Japanese name, *Watari*) under the guise of the edible mushroom *P. ostreatus*. Eventually, the old supervisor ate all of the mushroom dishes and said, “For years, this old priest has never had such deliciously cooked *watari”*; the old supervisor knew all along, but he was of a special constitution such that he never got affected by the toxin [69].

Currently, Japan experiences approximately 30 cases of mushroom poisoning annually, and the cases of *O. japonicus* are among the highest every year, accounting for approximately 50% of the cases [70]. The primary toxic substance was isolated and identified as illudin S (lampterol) by two Japanese organic chemists, Koji Nakanishi (1925–2019) and Takeshi Matsumoto (1923–2014) [71,72]. The major symptoms of the toxin are vomiting, diarrhea, and stomachache. In one case, curiously, “They felt dizzy and everything around them appeared blue to their eyes. Moreover, they experienced a feeling as if a number of fire-flies were flying around them” [41]. The draft genome sequence of this species (Korean cultivar) was assembled, and bioluminescence-related genes were identified [73]. Haneda reported weak luminescence of the spore mass on moist paper based on specimens collected from Akita Prefecture [30]. This species has an essential role in beech log decomposition in cool temperate forest floors in Japan [74], and because of the recent decline in natural beech forests, it is listed in some prefectural Red Data as a threatened species (e.g., Mie, Osaka, and Shimane Prefectures) [63]. *Tsukiyo-také* is one of the seasonal terms of the Japanese short poetry *Haiku* for mid-autumn [75].

*“Wolves wander along/mountain trails, their ways lit by/moonlit-night mushrooms”,* Kansuke Naka (1885–1965, a Japanese novelist, essayist, and poet) (translated by Nathaniel Guy [3], and his personal communication).

#### 5.1.3. Family **Physalacriaceae** (Figure 12)

***Armillaria cepistipes*** Velen.

Japanese name: *Kuroge-narataké* [55]

Remarks: Bioluminescence of Japanese samples (mycelium) was reported by Hiroi, 2006 [18]. Japanese *Kuroge* means black hair.

***Armillaria gallica*** Marxmuller & Romagn.

Japanese name: *Yawa-narataké* or *Watage-narataké* [55]

Remarks: Bioluminescence of Japanese samples (mycelium) was reported by Hiroi, 2006 [18]. Luminescence of the rhizomorphs has been reported elsewhere [76] but not from the Japanese samples. The fruitbodies of several *Armillaria* species, including *A. gallica* and *A. mellea*, are popular in Japan as a tasty edible mushroom species [62]. Japanese *Narataké* means ‘oak mushroom’, although the *Armillaria* species also grow on other varieties of tree. *Watage* means fluff because the veil of this mushroom is covered by a fluff-like structure [21]. *Yawa* means soft.

***Armillaria mellea*** (Vahl) P. Kumm.

Japanese names: *Narataké, Harigane-také,* or *Kuri-také* [55]

Remarks: Bioluminescence of Japanese samples (mycelium), which are sometimes called *Armillaria mellea* subsp. *nipponica* J.Y. Cha & Igarashi, was reported by Hiroi, 2006 [18]. Luminescence of young rhizomorphs is also reported [62]. Japanese *Harigane* and *Kuri* mean wire and chestnut tree, respectively.

***Armillaria nabsnona*** T. J. Volk & Burds.

Japanese name: *Yachi-narataké* [55]

Remarks: Bioluminescence of Japanese samples (mycelium) was reported by Hiroi, 2006 [18]. Japanese *Yachi* means marsh land because this species appears in marsh areas [68].

***Armillaria ostoyae*** (Romagn.) Herink

Japanese name: *Oni-narataké* or *Tsuba-narataké* [55]

Remarks: Bioluminescence of Japanese samples (mycelium) was reported by Hiroi, 2006 [18]. Japanese *Oni* and *Tsuba* mean a *Y*ō*kai* ogre and mushroom ring (annulus) [58]. The mushroom possesses an obvious veil. Rough scales on the cap evoke the image of violent *Oni* [21].

***Armillaria*** sp.

Japanese name: *Kitsubu-narataké* [77]

Remarks: Its taxonomic status has not been thoroughly studied, but it presumably represents a new species of the genus based on several morphological characteristics. Bioluminescence of Japanese samples (mycelium) was reported by Hiroi, 2006 [18]. In Japanese, *kitsubu* means yellow dots, referring to this characteristic of the cap surface.

***Desarmillaria tabescens*** (Scop.) R. A. Koch & Aime

Syn. *Armillaria tabescens* (Scop.) Emel

Japanese name: *Narataké-modoki* [55]

Remarks: Luminescence intensities of the fruitbody measured by a chemiluminescence detector largely depend on the specimens, but even in the most luminescent specimen, the light was too weak to be observed by the human eye [18,78]. Luminescent intensities of the mycelia also vary, but some could be clearly observed by the human eye [18,78]. The luminescence intensities are correlated with the strains of fruitbody and mycelium, suggesting that the luminescence characteristics are hereditary [18,78]. The species has long been known as *Armillaria tabescens* but was recently transferred to a newly established genus, *Desarmillaria* [79]. Japanese *-modoki* means pseudo, because this mushroom is similar to that of *Narataké* (*A. mellea*), but it possesses no veil [21]. This mushroom is regarded as edible but can cause gastrointestinal disorders when consumed in large quantities [10].

#### 5.1.4. Family **Pleurotaceae**

***Pleurotus nitidus*** Har. Takah. & Taneyama (Figure 13)

Japanese name: *Shiro-hikari-také* [47]

Remarks: The Japanese *Shiro-hikari-také* means ‘white luminescent mushroom’. This species from Ishigaki and Iriomote Islands, southern Japan, was described as being new [47]. However, it probably needs to be transferred to other genera containing bioluminescent species, such as *Neonothopanus* or *Nothopanus,* based on its morphological characteristics. Currently, no bioluminescent species are known from the genus *Pleurotus* and closely related genera. The only exception can be seen in *Pleurotus eugrammus* [46], but it is now treated as *Nothopanus eugrammus*, a species more closely related to *Omphalotus* and only distantly related to *Pleurotus* [80].

### 5.2. Nonbioluminescent Species Based on Samples Collected from Japan

***Panellus stipticus*** (Bull.) P. Karst.

Japanese name: *Wasabi-také* or *Himé-kawaki-také* [55]

Remarks: The Japanese name is *Wasabi-také* because of the strong pungent taste of the fruitbody, similar to ‘wasabi’, a spicy green paste served with sushi [31,62]. It is widely distributed in the world [60,62]. The North American population of this species is luminescent, but the European and Japanese strains are nonluminescent. The North American and European strains are interfertile, and luminosity is dominant over nonluminosity [81]. Samples from Turkey (nonbioluminescent) lack the genes related to bioluminescence (luciferase, hispidin-3-hydroxylase, and hispidin synthetase) in the genome [50]. Fruitbodies are frequently attacked by slugs, which may be important agents in the dispersal of their spores [82], but their involvement in bioluminescence for the attraction of dispersers is unknown (see Section 7). Japanese *Kawaki-* means ‘dried-’.

### 5.3. Potentially Bioluminescent Species in Japan

There are several fungal species that have been reported to be bioluminescent elsewhere but not in Japan. Some (probably most) of these species are bioluminescent at least in mycelial stages. According to Desjardin et al., “mycelium of most (if not all) *Armillaria* species is luminescent” [46], and thus the *Armillaria* species, in which bioluminescence has not been reported in Japan, might also be bioluminescent: for example, *A. jezoensis* Cha & Igarashi (Japanese name, *Kobari-narataké* [55]), *A. singula* J. Y. Cha & Igarashi (*Hitori-narataké* [55]), and *A. tympanitica* (Berk. & M. A. Curtis) Sacc. (which has no Japanese name, but was collected once from Bonin Isl. [83], although Ito suggested its species identification was doubtful [84]). Since bioluminescence of *Gerronema viridilucens* mycelia and fruitbodies has recently been reported from Brazil [85], the congeneric species recorded from Japan (such as *G. holochlorum* and *G. nemorale*) may also be bioluminescent. Table 1 summarizes the species reported to be bioluminescent elsewhere but not in Japan.

## 6. Bioluminescence

### 6.1. Before the Meiji Period

Chemical research on luminous fungi began in the 17th century, coinciding with the Scientific Revolution. Sir Robert Boyle (1627–1691), a founder of modern chemistry, examined luminous mycelia on rotten wood, called ‘shining wood’, using his newly invented vacuum pump and showed the involvement of air in luminescence [98]. Although Boyle did not know that the luminescence of ‘shining wood’ was caused by the mycelia of luminous fungi, his elegant experiments were recognized as a pioneering study leading the chemical understanding of bioluminescence. Jean-Henri Fabre (1823–1915), a French entomologist, demonstrated the requirement of oxygen for the luminescence of the ‘jack-o’-lantern’ mushroom growing on the olive tree, *Omphalotus olearius* [99]. In contrast, Japan had a late start in modern science, which is also true for the chemistry of fungal bioluminescence. In the days of Boyle and Fabre, during Japan’s Edo Period, Japanese people still believed in the story of *Yamiyo-také* (see above) or simply composed *Haiku* poems on the luminescence of mushrooms [100].

### 6.2. After the Meiji Revolution

In Japan, modern science started after the Meiji Revolution in 1868. Most likely, S. Kawamura was the first Japanese individual who studied the bioluminescence of fungi. In his descriptive paper of *O. japonicus*, he also reported the effects of nitrogen, hydrogen, and oxygen gasses on the luminosity of *O. japonicus* fruitbodies, showing that the luminescence did not fade when it was treated with oxygen [41,101]. These results are basically the same as those using *O. olearius* by Fabre [99], as described above. Kawamura also observed that the juice squeezed out from the luminous gills had no luminosity [41,67].

### 6.3. Airth’s Achievement

Bioluminescence reactions in vitro using extracts of luminous fungi had not yet been achieved despite decades of trials (Harvey, 1952 [24], and references therein), but Airth and McElroy finally succeeded using the extracts of ‘luminous fungi’ species [102]. They detected luminescence when a hot-water extract from the mycelia of the luminous fungus *Armillaria mellea* and a cold-water extract from the mycelia of the luminous fungus ‘*Collybia velutipes*’ were mixed in the presence of NADH or NADPH [102,103]. Airth and Foerster showed that this reaction consists of at least two steps, involving the reduction of unidentified dehydro- or oxyluciferin (luciferin precursor) by NADH or NADPH with a soluble enzyme (approximately 25 kDa [104]) and light-emitting oxidation of luciferin by molecular oxygen with an insoluble membrane-bound enzyme [103,105]. Airth and Foerster also showed that a cold-water extract from *Panellus stipticus* (*Panus stipticus*, in their paper) mycelia (luminescent strain) exhibited luminescence activity for a hot-water extract from *A. mellea* mycelia with NADH [106]. Based on these results, they suggested that fungal luminescence is explained as a luciferin–luciferase reaction, and the presence of reduced pyridine nucleotides in the reaction mixture is key to reproducing the luminescence in vitro [104]. Regarding the species ‘*Collybia velutipes*’, the authors of this paper wrote that the mycelia of ‘*C. velutipes*’ used for cold-water extraction was “luminous” [106], but this species is currently recognized as nonluminous *Flammulina velutipes* [46,76]. We confirmed that a cold-water extract of the *F. velutipes* (or *F. filiformis*) fruitbody has no luminescence activity by mixing a hot-water extract of the *M. chlorophos* fruitbody and NADPH. Interestingly, on the other hand, strong luminescence activity was detected in a hot-water extract of the *F. velutipes* fruitbody by mixing a cold-water extract of the *M. chlorophos* fruitbody and NADPH [107]. Accordingly, we conclude that Airth’s research group misidentified unknown luminous mycelia as ‘*C. velutipes*’, and true *F. velutipes* has no enzymatic activity but unexpectedly contains a luciferin precursor. Later, we realized that several nonluminous fungi contain a large amount of the luciferin precursor hispidin [108], as described below.

### 6.4. Various Candidate Compounds

Candidates of luciferin or compounds involved in fungal luminescence have been proposed by several research groups (Figure 14). Airth et al. [103] showed that the luciferin precursor is soluble in phosphate buffer (pH 7.5) with 2% Tween 80 rather than in organic solvents, including acetone, chloroform, benzene, and diethyl ether [103]. In 1966, Kuwabara and Wassink reported the purification and crystallization of an ‘active substance’ (luciferin or dehydroluciferin) from the mycelia of the luminous fungi *Mycena citricolor* (*Omphalia flavida*, in the paper); this substance had enzymatic luminescent activity for “Dr. Airth’s fungal luciferase system” and nonenzymatic chemiluminescent activity in the presence of H_2_O_2_ [109]. The crystal was a brownish-orange ‘microcrystalline solid’ (needle-like crystalline, see Kuwabara & Wassink, 1966 [109]). The UV, IR, and fluorescence of this ‘active substance’ were measured, but the chemical structure was not determined. Later, Airth himself examined the enzymatic activity of this crystal shipped from Kuwabara using a cold-water extract of ‘*C. velutipes*’ with NADH or NADPH. However, the result was negative: “Since the possibility of inactivation during transport has not been disproved, the question as to whether Kuwabara and Wassink (1966) did crystallize fungal luciferin remains unanswered” [104]. Wassink suggested that there remains some doubt whether this crystallized compound was indeed fungal luciferin [76]. From the current viewpoint, it is believed that the basic optical properties (UV and fluorescent spectra) of the ‘active substance’ correspond with neither of the candidate precursors of luciferin (hispidin and caffeic acid); both have luminescence activities to the crude buffer extract of the luminescent fungi [108,110]. Of note, Seishi Kuwabara, the first author of the paper about crystallization of the ‘active substance’, is a Japanese biologist. His biography is uncertain, but he also worked on bacterial luciferase with a bioluminescence researcher, Milton Cormier, in the 1960s (see Kuwabara et al., 1965 [111]).

Nakanishi and his colleague isolated fluorescent compounds illudin S (Figure 14A) and ergosta-4,6,8(14),22-tetraen-3-on (Figure 14B), which showed emission maxima close to those of fungal luminescence, from the mycelia and fruitbody of *O. japonicus* as potential substances involved in the bioluminescence of fungi [112]. Illudin S is a compound responsible for the toxicity of *O. japonicus,* as shown above. The bioluminescence and chemiluminescence activities of these compounds have not been examined.

Lampteroflavin (Figure 14C) is a pentofuranosyl riboflavin compound isolated from *O. japonicus* that possesses green fluorescence (λmax = 524 nm) identical to the bioluminescence spectrum [113,114]. Minoru Isobe, a Japanese organic chemist, and his colleagues suggested that this compound is the light emitter of *O. japonicus* luminescence because it was the only fluorescent compound that showed an identical fluorescence spectrum to the in vivo bioluminescence detected in the lamellae of this mushroom [114]. Several chemicals, including flavins, exhibit chemiluminescence in Fenton’s reagent (Fe^2+^ and H_2_O_2_) [115], and Isobe showed that the chemiluminescence of lampteroflavin in Fenton’s reagent was significantly stronger than that of other flavins (lumiflavin, riboflavin, FMN, and FAD). Based on this result, Isobe expected that lampteroflavin would also be involved in the luminescence reaction itself [116]. On the other hand, O’Kane et al. suggested that flavin could not be the light emitter because luminous mushrooms are pigmented, and the ‘corrected’ bioluminescence spectrum of *O. japonicus* mycelia, which do not have the filter effect by pigmentation, did not match the fluorescence spectrum of flavin [117].

Nobel Prize winner Osamu Shimomura (1929–2018), a Japanese chemist/biochemist living in the USA, also contributed to research on fungal bioluminescence. He won the prize by finding the green fluorescent protein from luminous jellyfish *Aequorea aequorea*, and he also studied fungal bioluminescence using a *Panellus stipticus*; he was fascinated by this luminous mushroom, which appeared on an oak stump cut for building his new house at Falmouth in the 1980s [118]. He cultivated the fruitbody himself by applying his *Shii-také* (*L. edodes*) cultivation experience [119,120]. Using those cultivated *P. stipticus* mushrooms, Shimomura isolated panal, a sesquiterpene aldehyde, and its derivatives (Figure 14D), PS-A and PS-B, as possible candidates for luciferin precursors from the fruitbody [121,122,123]. After activation with primary amine, these panal derivatives exhibited chemiluminescence in the presence of Fe^2+^ and H_2_O_2_. The emission spectral peak of the chemiluminescence depends on the reaction conditions, some of which were close to the luminescence spectra in vivo. The same chemiluminescence was also observed for various other luminous fungi, such as *Armillaria* mycelia and *Mycena* fruitbodies [124]. Based on these results, Shimomura suggested that the bioluminescence of fungi might be a nonenzymatic reaction, which is inconsistent with Airth’s suggestion.

Other candidate compounds for luciferin or its precursor of fungal luminescence were isolated from the mycelia of *M. citricolor* by O. Shimomura (see Shimomura, 2006 [119]), and the chemical structure was analyzed by a Japanese organic chemist, Hideshi Nakamura (1952–2000). He suggested that all luciferin precursors contain a common catechol-derived group (4-(3,4-dihydroxyphenyl)-3-buten-2-carbonyl group) (Figure 14E) in their structures (personal communication from Dr. H. Nakamura, 1998, as referred to in Shimomura, 2006 [119]). To our surprise, this partial structure and UV absorption peak of the luciferin precursor (369 nm; see Shimomura, 2006 [119]) match those of hispidin, which we currently think is a true luciferin precursor of fungal bioluminescence [108].

### 6.5. Current Hispidin-Recycling Mechanism

The enzymatic reaction proposed by Airth’s group has been verified using several species of luminous fungal fruitbodies and mycelia, including the genera representing all four bioluminescent fungal lineages, *Gerronema*, *Mycena, Armillaria,* and *Neonothopanus* [125,126,127,128]. Oliveira et al. also examined cross-reactions between species, including the nonluminous fungi *Filoboletus gracilis* and *Mycena singeri*, and the results were all negative when nonluminous species were used for the assay [128]. Based on these results, they suggested that all luminous fungi share the same or similar luciferin and luciferase, whereas these components are absent in nonluminous species [128].

The luminescence mechanism of Japanese *M. chlorophos* was recently investigated by two independent Japanese research groups [129,130]. Mori et al. showed that Dubois’ classical luciferin–luciferase test was negative, bioluminescent components are insoluble in various surfactants, and the luminescence is heat unstable and thus will be an enzymatic reaction [129]. Hayashi et al. isolated an unknown compound that exhibited a UV absorption spectrum similar to that of flavins from the fruitbody, and the fluorescence peak almost matched both riboflavin and the bioluminescence of *M. chlorophos* [130]. Based on these findings, the authors concluded that this flavin-like compound is a factor in fungal bioluminescence. The bioluminescence or chemiluminescence activity of this compound was not examined in this report.

In 2015, we eventually identified the chemical structure of fungal luciferin and its precursor as 3-hydroxyhispidin and hispidin (Figure 14F), respectively, using the mycelium of the Vietnamese luminous fungus *Neonothopanus nambi* [108]. In 2017, we reported the whole mechanism of fungal bioluminescence by determining the chemical structure of the reaction product as caffeylpyruvic acid as a light emitter (Figure 14F) [131]. In 2017, we also determined the presence of hispidin as an active compound for the NADPH-dependent bioluminescence reaction in the fruitbodies of Japanese *M. chlorophos*, *O. japonicus*, and Brazilian *N. gardneri* [110]. The unopened ‘young’ fruitbody of *M. chlorophos* is nonluminescent, but it starts luminescence within seconds to the visible level when soaking with hispidin solution (Figure 15). This can be explained by the presence of luciferase but no luciferin precursor, hispidin, in the young fruitbody [110].

Katsunori Teranishi, a Japanese organic chemist, proposed a conflicting suggestion of light emitter compounds of luminescence in the *M. chlorophos* mushroom as riboflavin, riboflavin 5′-monophosphate, and/or flavin adenine dinucleotide [132]. He stated that 3-hydroxyhisidin can produce light by partially purified luciferase from *M. chlorophos*, but it remains unclear whether the compound actually produces light in the natural tissue of luminous fungi [133].

In 2018, we identified the luciferase gene, as well as luciferin-regenerating enzymes from *N. nambi* (Figure 14F); these genes were clustered on the fungal genome, and the molecular size of the luciferase (Luz) was approximately 28.5 kDa [50]. When these genes were transferred to yeast, the strain emitted the same green light. Currently, bioluminescent gene clusters have been identified in various luminous fungal species [50,51,73]. Furthermore, this fungal bioluminescence system is completely different from all known bioluminescent systems in other luminous organisms [134], suggesting that basic bioluminescence mechanisms are common in all luminous fungi; thus, bioluminescence in fungi has a single evolutionary origin [51]. In 2020, a self-luminescent plant was genetically produced using fungal luminescence genes [135]. Indeed, this bioengineering was realized because a common compound produced by all plants, caffeic acid (Figure 14F), is coincidentally a precursor of fungal luciferin. This finding has the potential to advance plant science as a novel tool for bioimaging technology. To date, no higher plant species with natural bioluminescence has ever been reported [23], but in the future, fungal luminescent genes may illuminate the city as glowing street trees [136].

## 7. Biological Function

For the biological function of fungal bioluminescence, many hypotheses have been proposed [137] but are not yet conclusive [138]. Arguably the most plausible hypothesis is that light attracts insects as a spore vector [139]. In Japan, we observed various animals visiting the fruitbodies of *M. chlorophos* (Figure 16) at night. Undescribed Japanese species *Roridomyces* sp. (*Aya-hikari-také*) has luminescent spores, suggesting the presence of a spore dispersal function by light-attracted animals [10]. Aposematism is another strong hypothesis [137]. However, for example, *O. japonicus* is toxic to humans, but many insect species visit and consume the fruitbody [140] (Figure 16).

## 8. Conclusions

In this review, we demonstrated the enthusiasm for luminous fungi in Japan, from old folklore and taxonomic surveys to current popular entertainment and life sciences. Luminous mushrooms activate ecotourism, and the biotechnology born from the science of luminous fungi has the potential to change our urban life.

## Figures and Tables

**Figure 1 jof-09-00615-f001:**
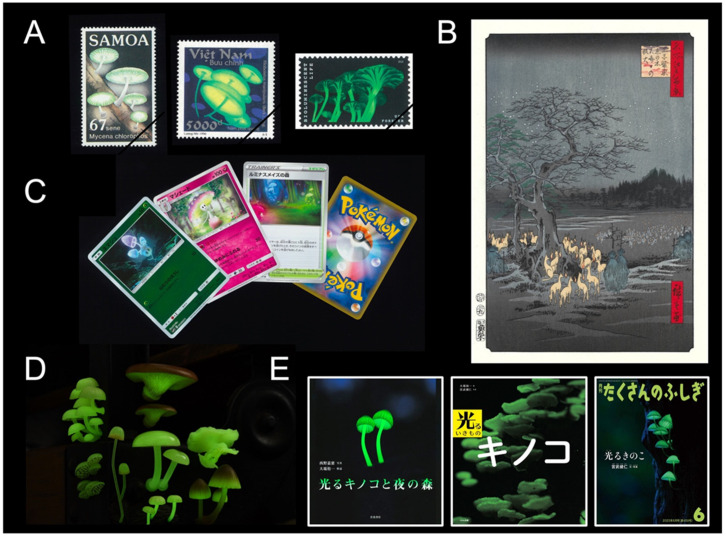
(**A**) Luminous mushroom stamps. (From left to right) *Mycena chlorophos/*West Samoa, 1985; *Mycena manipularis/*Việt Nam, 1996; *Mycena lucentipes/*USA, 2018. There are no luminous mushroom stamps in other countries, including Japan. (**B**) Foxfire at Ōji (*Ōji Sh*ō*zoku Enoki Ōmisoka no Kitsune-bi*) by Hiroshige Utagawa (recarved edition, original print in 1857). (**C**) Pokémon cards. English names (from left to right): Morelull (basic), Shinotic (stage 1), and Glimwood Tangle (stadium), produced by The Pokémon Company (Tokyo, Japan). (**D**) Capsule toys: “luminous mushroom magnet” series (2015-) of eight Japanese luminous mushroom species, produced by Ikimon Co. (Tokyo, Japan). The diameter of the model of *Mycena chlorophos* (center) is approximately 30 mm, which is close in size to the largest specimens found in the wild. (**E**) Picture books of luminous mushrooms. (From left to right) Nishino & Oba, 2013; Oba & Miyatake, 2015; Miyatake, 2023. All are Yuichi Oba’s personal collections.

**Figure 2 jof-09-00615-f002:**
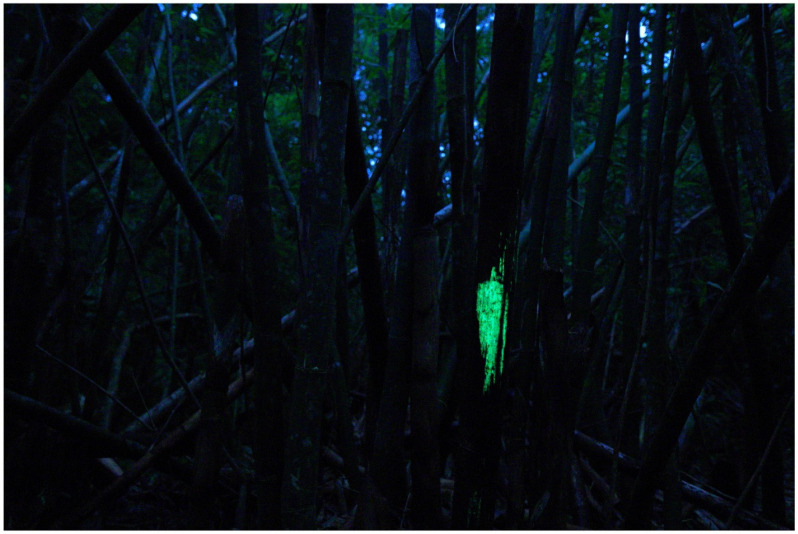
Mycelia of *Mycena stellaris* growing on bamboo. Photo by Yoshinori Nishino on Ishigaki Isl., Okinawa Prefecture.

**Figure 3 jof-09-00615-f003:**
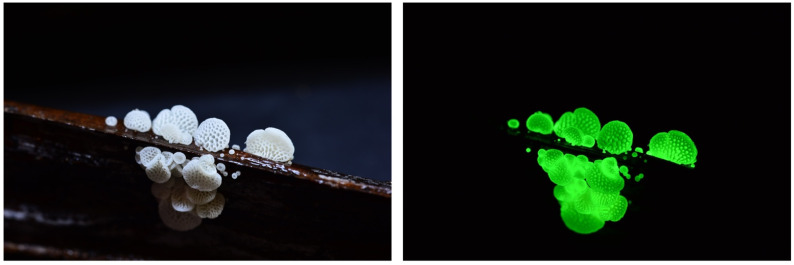
Fruitbody of *Favolaschia peziziformis.* Photo by So Yamashita on Hachijo Island, Tokyo.

**Figure 4 jof-09-00615-f004:**
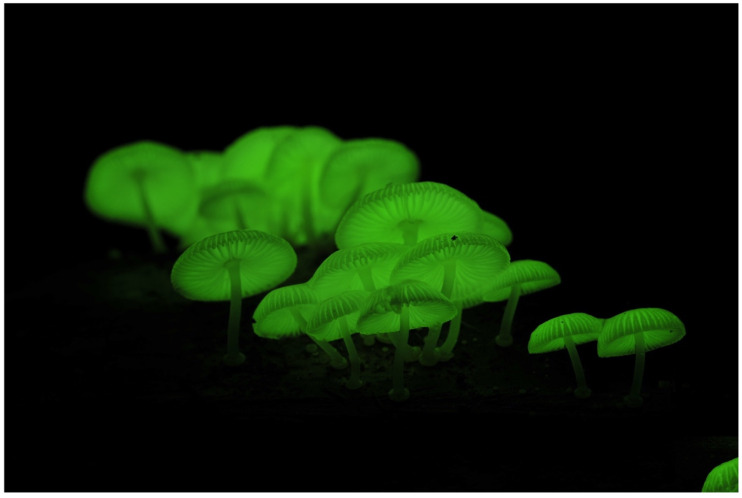
Fruitbody of *Mycena chlorophos*. Photo by So Yamashita on Hachijo Island, Tokyo.

**Figure 5 jof-09-00615-f005:**
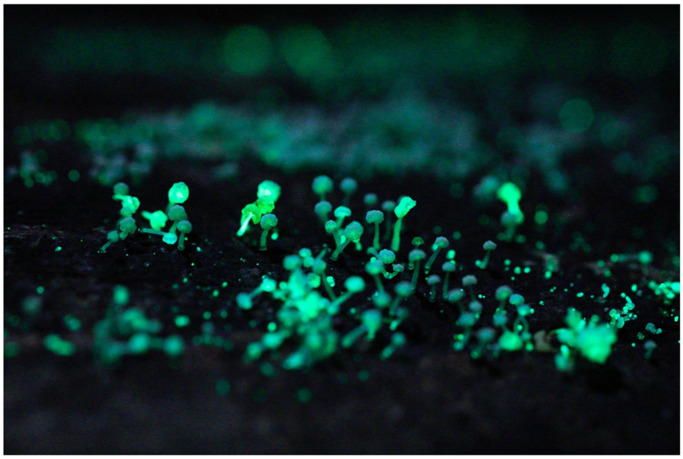
Fruitbody of *Mycena lazulina*. Photo by Yoshinori Nishino on Iriomote Island, Okinawa Prefecture.

**Figure 6 jof-09-00615-f006:**
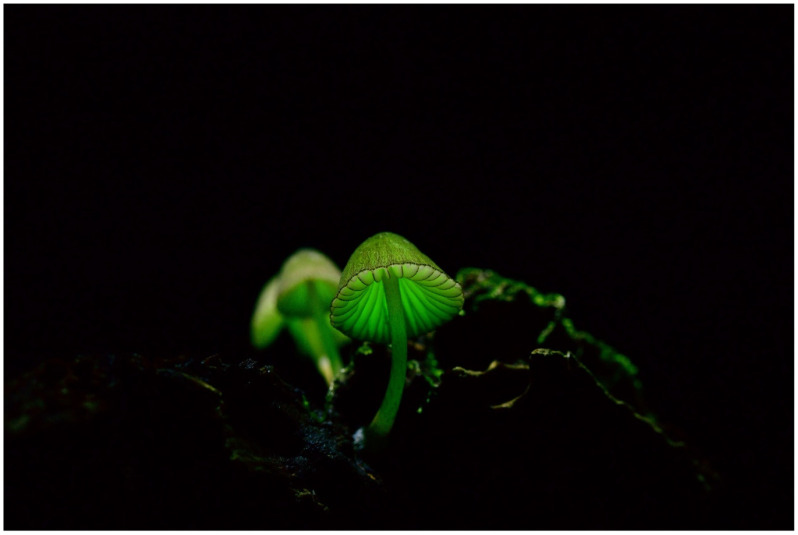
Fruitbody of *Mycena lux-coeli*. Photo by So Yamashita on Hachijo Island, Tokyo.

**Figure 7 jof-09-00615-f007:**
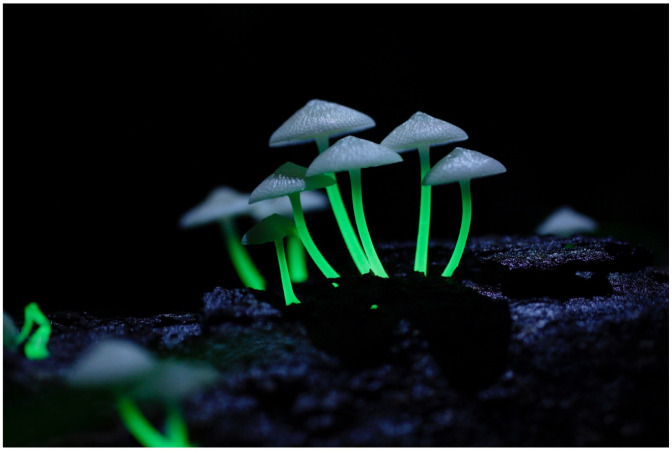
Fruitbody of *Mycena manipularis*. Photo by Yoshinori Nishino on Ishigaki Island, Okinawa Prefecture.

**Figure 8 jof-09-00615-f008:**
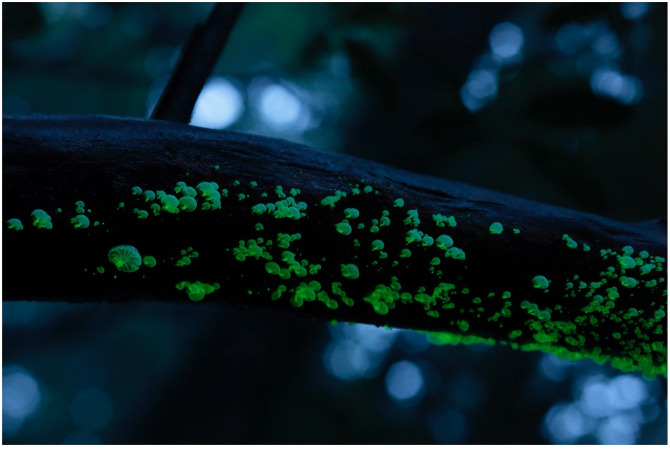
Fruitbody of *Mycena stellaris*. Photo by Yoshinori Nishino at Kunigami, Okinawa Isl., Okinawa Prefecture.

**Figure 9 jof-09-00615-f009:**
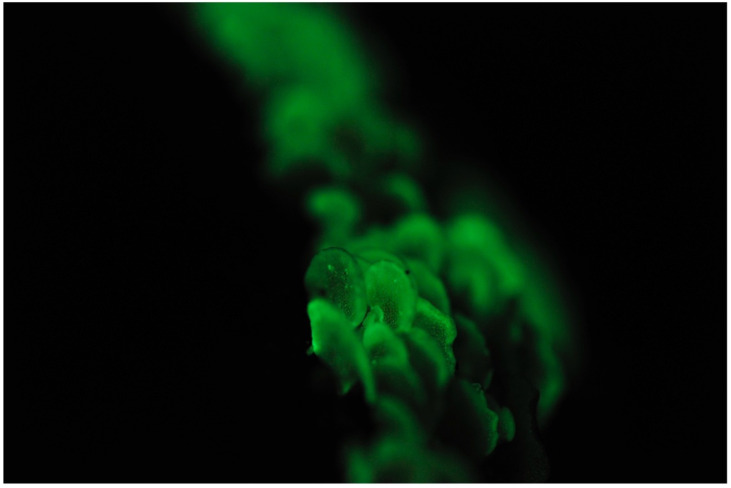
Fruitbody of *Panellus pusillus.* Photo by So Yamashita on Hachijo Island, Tokyo.

**Figure 10 jof-09-00615-f010:**
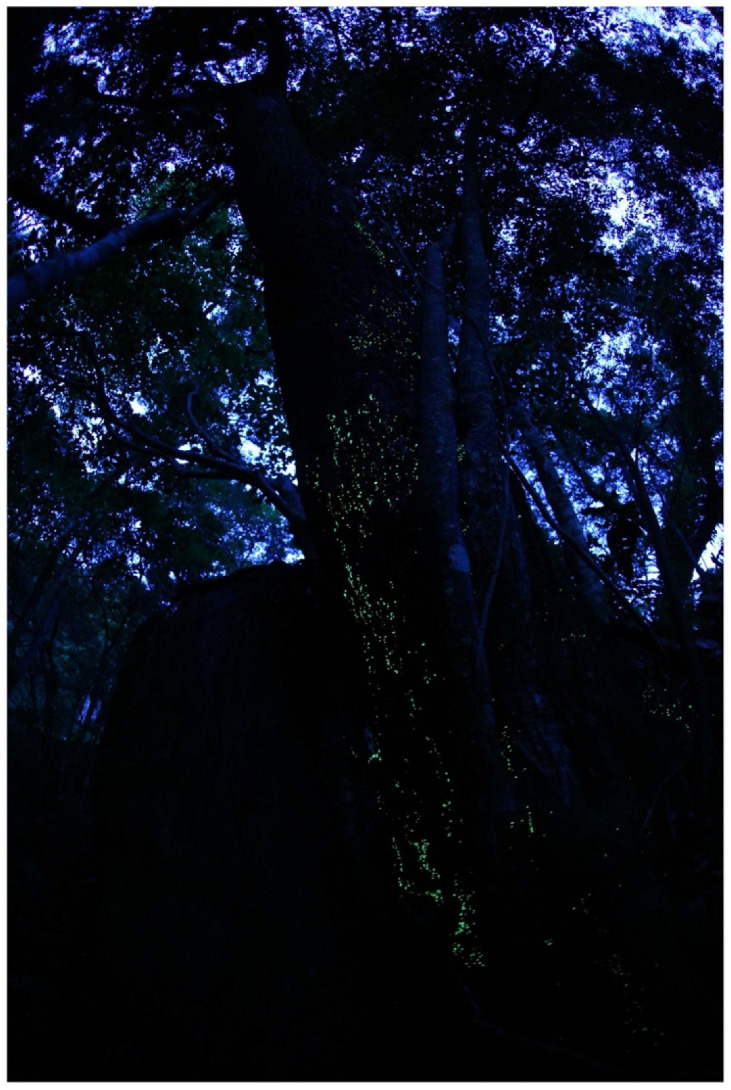
Fruitbody of *Resiomycena fulgens.* Photo by Takehito Miyatake on Hachijo Island, Tokyo.

**Figure 11 jof-09-00615-f011:**
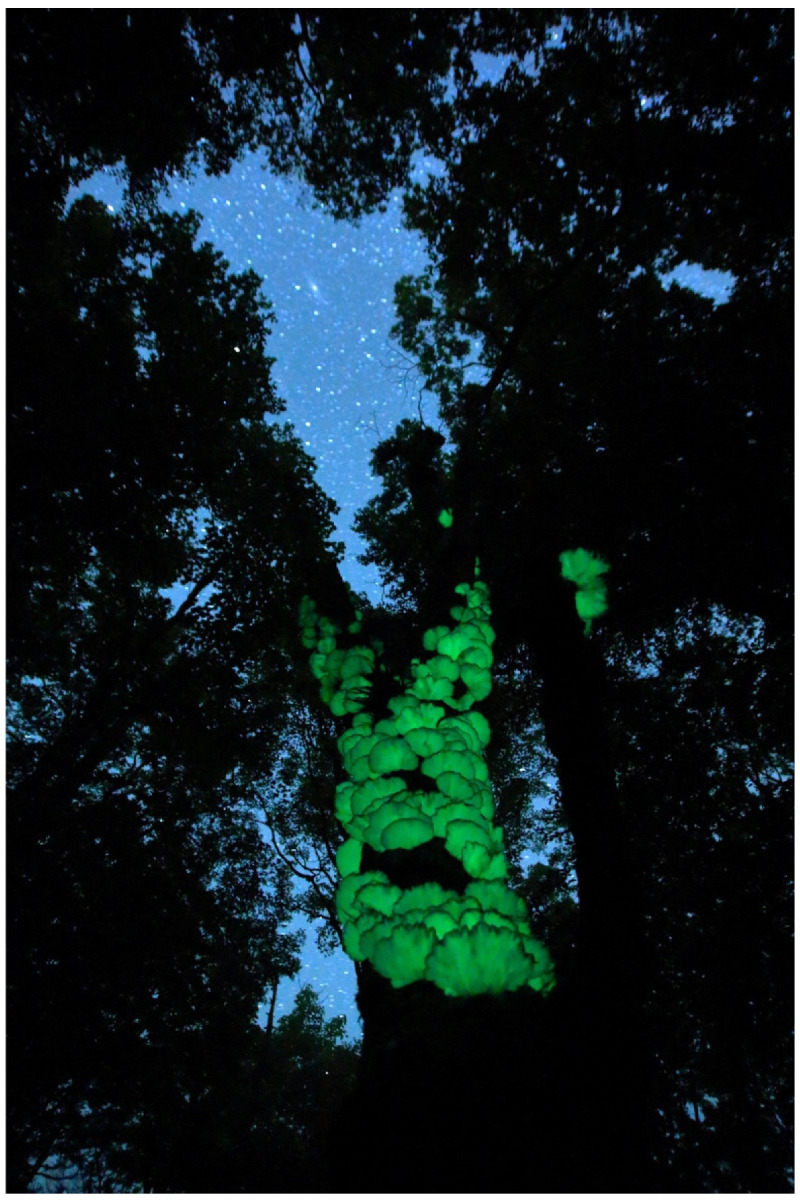
Fruitbody of *Omphalotus japonicus*. Photo by Yoshinori Nishino at Mt. Odaigahara, Nara Prefecture.

**Figure 12 jof-09-00615-f012:**
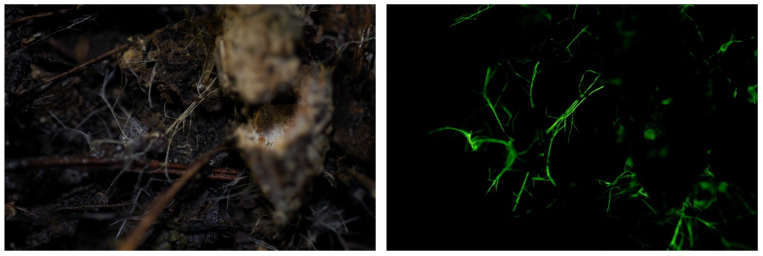
Rhizomorphs of *Armillaria* sp. Photo by So Yamashita on Hachijo Island, Tokyo.

**Figure 13 jof-09-00615-f013:**
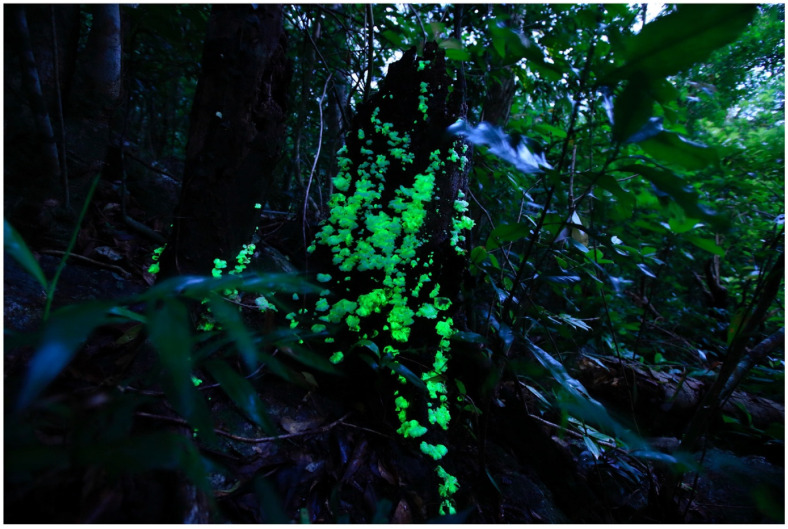
*Pleurotus nitidus.* Photo by Yoshinori Nishino on Ishigaki Island, Okinawa Prefecture.

**Figure 14 jof-09-00615-f014:**
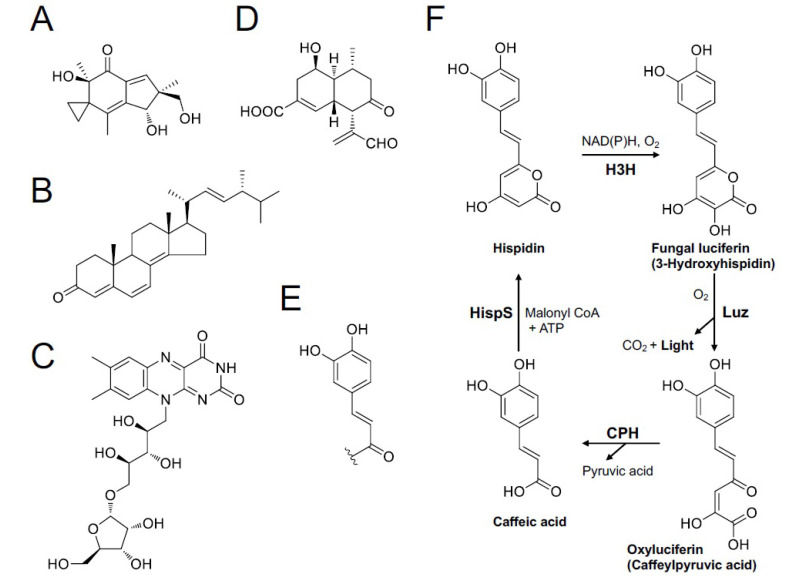
Chemical structure of luciferin candidates. (**A**) Illudin S, (**B**) ergosta-4,6,8(14),22-tetraen-3-on, and (**C**) lampteroflavin from *Omphalotus japonicus*. (**D**) Panal from *Panellus stipticus*. (**E**) A common partial structure of the luciferin precursors from *Mycena citricolor*. (**F**) Scheme of hispidin-recycling reaction determined in *Neonothopanus nambi*. Four enzymes involved in the cycle are shown in bold: H3H, hispidin-3-hydroxylase; Luz, fungal luciferase; CPH, caffeylpyruvate hydrolase; and HispS, hispidin synthase.

**Figure 15 jof-09-00615-f015:**
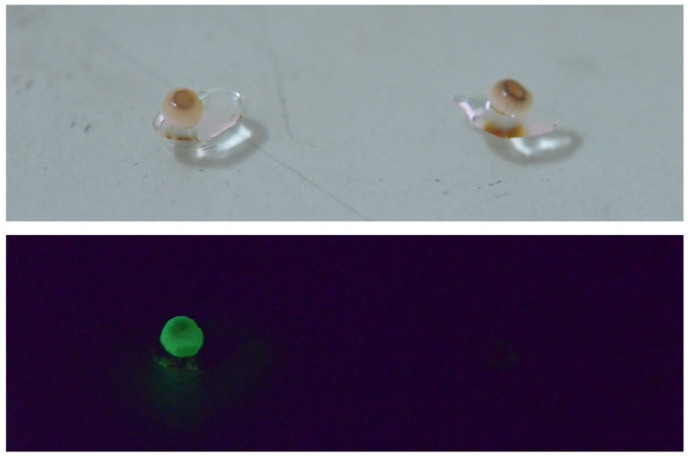
Chemical induction of the luminescence of the unopened young *Mycena chlorophos* fruitbody (5 mm diameter) by a buffer with and without 220 μM hispidin (**left** and **right**). Photos were taken 10 s after treatment.

**Figure 16 jof-09-00615-f016:**
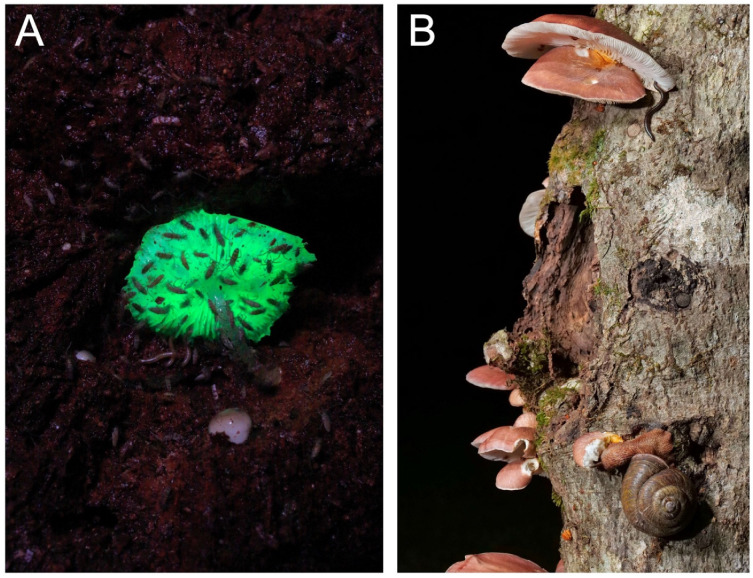
Fruitbodies of *Mycena chlorophos* with collembolas gathering under the gills (**A**) and *Omphalotus japonicus* with millipede and land snail grazing (**B**). Photos by Yoshinori Nishino on Bonin Isls., Tokyo (**A**) and by Takehito Miyatake at Tokushima Prefecture (**B**).

**Table 1 jof-09-00615-t001:** Potentially bioluminescent species in Japan.

Taxon	Japanese Name (*1)	Bioluminescence (References) (*2)
Family Mycenaceae		
*Mycena epipterygia* (Scop.) S.F. Gray	*Nameashi-také*	Mycelium (Bothe, 1931 [86]; Wassink, 1978 [76], 1979 [45]; Desjardin et al., 2008 [46])
*Mycena galopus* (Pers.) P. Kumm.	*Nise-chishio-také*	Mycelium (Bothe, 1931 [86]; Berliner, 1961 [87]; Wassink, 1978 [76], 1979 [45]; Treu & Agerer, 1990 [88]; Desjardin et al., 2008 [46])
*Mycena haematopus* (Pers.) P. Kumm.	*Chishio-také*	Mycelium (Treu & Agerer, 1990 [88]; Bermudes et al., 1992 [89]; Desjardin et al., 2008 [46]): Basidiomes (weak) (Bermudes et al., 1992 [89]; Desjardin et al., 2008 [46])
*Mycena inclinata* (Fr.) Quél.	*Sembon-ashinaga-také*	Mycelium (Wassink, 1978 [76], 1979 [45]; Desjardin et al., 2008 [46])
*Mycena olivaceomarginata* (Massee) Massee	*Fuchidori-kunugitaké* (*3)	Mycelium (Wassink, 1978 [76], 1979 [45]; Desjardin et al., 2008 [46])
*Mycena pura* (Pers.) P. Kumm. (*4)	*Sakura-také*	Mycelium (Treu & Agerer, 1990 [88]; Desjardin et al., 2008 [46]): gill of basidiome (Bothe, 1931 [86]; Wassink, 1978 [76], 1979 [45])
*Mycena rosea* (Bull.) Gramberg	*Sakurairo-také* [68]	Mycelium (Treu & Agerer, 1990 [88]; Desjardin et al., 2008 [46])
*Mycena sanguinolenta* (Alb. & Schwein.) P. Kumm.	*Himé-chishio-také*	Mycelium (Bothe, 1931 [86]; Wassink, 1978 [76], 1979 [45]; Desjardin et al., 2008 [46])
*Mycena stylobates* (Pers.) P. Kumm.	*Kyūban-také*	Mycelium (Bothe, 1931 [86]; Wassink, 1978 [76], 1979 [45]; Desjardin et al., 2008 [46])
*Roridomyces roridus* (Fr.) Rexer	*Nunawa-také*	Mycelium (Josserand, 1953 [90]; Wassink, 1978 [76], 1979 [45]; Desjardin et al., 2008 [46])
Family Physalacriaceae		
*Armillaria fuscipes* Petch (*5)	*Ashiguro-narataké* [91]	Mycelium (Wassink, 1978 [76], 1979 [45]; Berliner, 1961 [87]; Desjardin et al., 2008 [46]): Rhizomorph (Wassink, 1978 [76])
*Armillaria sinapina* Berube & Dessur.	*Hotei-narataké*	Mycelium (Mihail, 2015 [92])
*Desarmillaria ectypa* (Fr.) R.A. Koch & Aime	*Yachihiro-hidataké*	Mycelium, rhizomorph, basidiomes (Ainsworth, 2004 [93])

*1. The Japanese names were adopted from the list by Katumoto, 2010 [55], unless they have a more recent name. *2. Question marks represent the references showing that luminescence is doubtful or worth further investigation. Wassink (1948) [94] was not referenced in this list because his recent review papers [45,76] are considered updated versions of it. *3. Hongo (1989) [95] suggested that *Mycena neoavenacea* may be the same species as *Mycena olivaceomarginata*. *4. Molecular analysis suggests that the current *M. pura* morphospecies represent the species complex [96], and the bioluminescent ability of each phylospecies is unknown. *5. The morphological characteristics of this mushroom (named *Ashiguro-narataké* in Japanese, from Amami-Oshima) appeared indistinguishable from those of *A. fuscipes*, but the species name was not confirmed [97].

## Data Availability

Not applicable.

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
