# Peer review of "The Luminous Fungi of Japan"

_jof, 2023, doi:10.3390/jof9060615_

Round 1

Reviewer 1 Report

The manuscript presents a review of the bioluminescent fungi of Japan focusing on multiple aspects: from legends and folklore to the taxonomy and modern biochemistry of the phenomena. There is a lot of information about bioluminescent fungi species described from various part of Japan and the history of their studies. Of note is that many of the source references are in Japanese meaning they are not easily accessible for the general international readers which makes the review very wholesome.

While the present version of the manuscript has good readability, it can be improved by careful revision of the text. Below you can find the list of at least several sentences that should be revised.

Line 11 – “looking to find” should be changed to “looking for”. The sentence itself should be revised, it seems to lack some connection here “species richness is arguably thanks to the abundant presence of mycophiles, <never-ending interest of the scientists/public always> looking for new mushroom species and the tradition of night-time activities, such as firefly watching, in Japan.”

Line 63 – the sentence is confusing, please rephrase. Was it meant as “it is very interesting coincidence that old Japanese called an unknown glow on the ground “fox’s fire”, using the same metaphor as Europeans calling the glows “foxfire”? It is probably wise to mention here that the "fox" in European foxfire may derive from the Old French word faux, meaning "false", rather than from the name of the animal.

Line 66 – “called as”, should be either “called” or “known as”

Line 63-67 – in the present sentence “the yokai is responsible for the luminous mycelia growing”, but it should be “mycelia is responsible for what was considered to be the yokai glow”, please revise. Maybe the sentence itself is too big and it is better to separate the explanation about Kitsune-bi and Mino-bi as independent sentences.

Line 68 – “work by” should be changed to “work of”

Line 73-75 – “phenomena”: the word is used three times in a row, please rephrase

Line 124 – “the luminescence phenomena of mushrooms” should be changed to “the phenomenon of fungi bioluminescence”

Line 126 – the word “phenomena” is used again too early. Worldwide success?

Line 159 – “that was in 1915 by a mycologist” should be changed to “that was done in 1915 by a mycologist”

Line 199 – “luminescence … collected in Fukuoka”, please revise the sentence. Luminescence recorded/registered in Fukuoka?

Line 324 – italic for the names in Japanese

Line 326 – explanation about “take” should be placed in line 197 where it was mentioned for the first time in the list

Line 373, 382, 393, 431 – italic for the names in Japanese

Line 418 – bold

Overall, the presented manuscript is an interesting review and well deserved to be published in the Journal of Fungi after some minor text revision.

While the present version of the manuscript has good readability and quality of English, it can be improved by careful revision of the text.

Reviewer 2 Report

The paper submitted by Yuichi Oba and Kentaro Hosaka reviews the luminous fungi of Japan and their related topics including myth, taxonomy, and recent progress on bioluminescent fungi. In particular, plenty of photographs about luminous fungi are showed in this article. This paper is useful for researchers who are interested in the bioluminescence fungi and their application. However, some issues about the description and structure (see comments below) need to be addressed. As an overall impression, publication of this work can be considered after addressing major concerns.

1. Authors described in detail the taxonomy of luminous fungi in the section of taxonomy. My suggestion is that a figure like an evolutionary tree should be drawn to straightforward display the taxonomy of luminous fungi. I think this will make this section clearer and increase the readability.

2. Considering that the modern bioluminescent research on fungi is the important part involved in this paper, illustrations (schemes) about bioluminescence mechanism, or gene cluster of luciferase and luciferin, or luciferin biosynthesis, or hypothesis, or bioluminescent characteristic about luminous fungi, etc., should be provided to enrich the section of bioluminescence.

3. It will be better to represent the future directions of the research on luminous fungi in the section of conclusion, which will deepen this section.

4. The “figure 2” in line 583 seemed to be wrongly quoted. Please check and revise.

5. There is no table 1 in manuscript. Please check all tables.

6. Line 455, Table 2, “Bioluminescence (references) (*3)”, the annotation-symbol of *3 seemed to be wrongly labeled.

Reviewer 3 Report

The manuscript is a review of different fungal bioluminescent organism found in Japan. The review is very interesting especially because it brings information from Japanese literature in English. The manuscript contains many figures with single pictures. It would be easier to read if the pictures are put together. Furthermore, regarding the section 6, I regret there is not chemical structure of the fungal luciferin or more information regarding luciferase. In this section it would be very interesting to compare the bioluminescence between fungus and other bioluminescent organism. Authors can have a look on this old but nice review (Wilson T, Hastings JW. Bioluminescence. Annu Rev Cell Dev Biol. 1998;14:197-230. doi: 10.1146/annurev.cellbio.14.1.197. PMID: 9891783).

Reviewer 4 Report

The manuscript focuses on luminous fungi in Japan, showcasing their mythological stories, species classification, and modern scientific research achievements. The manuscript is well-written and highly interesting, resembling more of a popular science article rather than a professional academic review. Therefore, it may not be suitable for publication in JOF. This is reflected in the manuscript's illustrations, where figures 1/2/4/6 showcase luminous fungi that appear in printed materials, while the rest are various morphological photos of different luminous fungi. As for the section on modern scientific research, there are no pictures (although not necessary) and it is more academic in nature, deserving greater attention. However, the authors only provide a simple list without depth.

There are too many Japanese language articles cited in the references, which may not be very user-friendly for non-Japanese language learners.

 The manuscript is well-written。

Round 2

Reviewer 2 Report

This paper can be accepted.

Reviewer 4 Report

The quality of this reworked edition has been improved somewhat, especially in the current version where Figures 1 and 14 are well integrated and summarised. However, this manuscript is more like a popular science article than an academic paper. As the manuscript concludes, "In this review, we demonstrated the enthusiasm for luminous fungi in Japan, from old folklore and taxonomic surveys to current Luminous mushrooms activate ecotourism, and the biotechnology born from the science of luminous fungi has the potential to change our urban life". potential to change our urban life".  Therefore, personally, I cannot accept such a more popular type of paper to be published in a professional mycological journal.

  • The manuscript is well written.